# Real-time monitoring of single ZTP riboswitches reveals a complex and kinetically controlled decision landscape

Boyang Hua [1,7], Christopher P. Jones[2,7], Jaba Mitra[3], Peter J. Murray[4], Rebecca Rosenthal[4], Adrian R. Ferré-D'Amaré[2,8 ✉] & Taekjip Ha [1,4,5,6,8 ✉]

RNAs begin to fold and function during transcription. Riboswitches undergo cotranscriptional switching in the context of transcription elongation, RNA folding, and ligand binding. To investigate how these processes jointly modulate the function of the folate stress-sensing *Fusobacterium ulcerans* ZTP riboswitch, we apply a single-molecule vectorial folding (VF) assay in which an engineered superhelicase Rep-X sequentially releases fluorescently labeled riboswitch RNA from a heteroduplex in a 5′-to-3′ direction, at ~60 nt s$^{-1}$ [comparable to the speed of bacterial RNA polymerase (RNAP)]. We demonstrate that the ZTP riboswitch is kinetically controlled and that its activation is favored by slower unwinding, strategic pausing between but not before key folding elements, or a weakened transcription terminator. Real-time single-molecule monitoring captures folding riboswitches in multiple states, including an intermediate responsible for delayed terminator formation. These results show how individual nascent RNAs occupy distinct channels within the folding landscape that controls the fate of the riboswitch.

[1] Department of Biophysics and Biophysical Chemistry, Johns Hopkins School of Medicine, Baltimore, MD 21205, USA. [2] Biochemistry and Biophysics Center, National Heart, Lung, and Blood Institute, National Institutes of Health, Bethesda, MD 20892, USA. [3] Department of Materials Science and Engineering, University of Illinois at Urbana-Champaign, Urbana, IL 61801, USA. [4] T. C. Jenkins Department of Biophysics, Johns Hopkins University, Baltimore, MD 21218, USA. [5] Department of Biomedical Engineering, Johns Hopkins University, Baltimore, MD 21218, USA. [6] Howard Hughes Medical Institute, Baltimore, MD 21205, USA. [7] These authors contributed equally: Boyang Hua, Christopher P. Jones. [8] These authors jointly supervised this work: Adrian R. Ferré-D'Amaré, Taekjip Ha. ✉email: adrian.ferre@nih.gov; tjha@jhu.edu

As gene expression regulatory elements, riboswitches are structurally divided into two domains—an aptamer domain sufficient for ligand binding and an expression platform regulating gene expression[1–3]. These two domains share an overlapping sequence making their formation mutually exclusive. Ligand binding stabilizes the aptamer and prevents the shared segment from adopting its alternative conformation in which it is part of the expression platform[1–3]. The *F. ulcerans* ZTP riboswitch regulates gene expression by binding to 5-aminoimidazole-4-carboxamide riboside 5′-monophosphate and triphosphate (ZMP and ZTP, respectively), which are elevated during folate stress[4–6]. In the absence of ZMP, a transcription termination hairpin or terminator (i.e., the expression platform) forms and terminates transcription before the gene coding region (Fig. 1a). In the presence of ZMP, the aptamer-containing conformation is stabilized by ligand binding, preventing terminator formation, and allowing transcription to complete.

From bulk biochemical studies, it has been proposed that several riboswitches function through a kinetically controlled mechanism. In this regime, cotranscriptional folding is thought to create a time window for aptamer domain folding and ligand binding, which spans from complete synthesis of the aptamer domain (i.e., the earliest time when ligands can bind) to the synthesis of enough of the terminator to interfere with binding[7–9]. As a result, the gene-regulatory outcome would depend on the relative rates of transcription elongation, RNA folding, and ligand binding. However, systematic investigation of such a kinetic control model is hampered by the lack of methods to vary these concurrent processes independently. As riboswitches have gained interest as drugs targets[10,11], understanding their mechanisms is a natural step to inform these pursuits.

Investigations of the kinetic control model are often preceded by the observation of a discrepancy between equilibrium ligand-binding measurements and cotranscriptional riboswitch activation measurements, suggestive of a non-equilibrium (kinetically controlled) process. This discrepancy is then further examined through monitoring ligand binding and RNA folding kinetics. For example, bulk approaches have made use of structural probing techniques, NMR spectroscopy, and fluorescently labeled ligands to measure the RNA folding and ligand binding kinetics for different lengths of riboswitch RNAs representing different points

in transcription[8,9,12]. More recently, chemical probing approaches were combined with high-throughput sequencing and used to examine riboswitch structure during roadblocked transcription to obtain more complete maps of RNA folding for all transcript lengths[13,14]. However, these approaches cannot measure real-time folding dynamics because RNA conformations are measured long after transcription is stalled by a blockade.

Using an elegant optical tweezer assay, Frieda and Block observed the folding process of an adenine riboswitch during transcription[15]. Supporting the kinetic control model, the adenine aptamer was shown to fold and bind adenine no more than once before the transcription decision, which is not at equilibrium. However, due to technical limitations, the roles of several important parameters, such as elongation speed and pausing, were not tested. In addition, both a drawback and an advantage of the optical tweezer assay are that tension is applied to either the nucleic acid substrates or RNAP, thus examining force-dependent perturbation of the system.

To date, experiments studying the kinetic control model have primarily used RNAP (i.e., T7 RNA polymerase or *Escherichia coli* RNAP holoenzyme). Assembly of RNAP on labeled nucleotide substrates enables the study of RNAP by single-molecule FRET[16,17] but restricts dye placement to the 5′ end of the RNA and requires partial transcription of the RNA in a halted transcription elongation complex, which could alter riboswitch folding through an alternative pathway. As RNAP pausing is an established modulator of riboswitch activation in many systems[9,16,18–21], halting transcription could similarly disrupt or facilitate folding. Another issue that arises when using RNAP to examine cotranscriptional RNA folding is that mutations to the DNA template, hence changing the RNA sequence, can affect both RNA folding and interactions between RNAP and nucleic acid substrates. For example, a mutation to the terminator hairpin can result in changes in both its folding and its interactions with RNAP such that a single readout of transcription termination efficiency cannot determine if the mutation acts through changes in RNA folding or RNAP interaction.

To overcome the experimental shortcomings of employing RNAP, we seek an alternative for examining cotranscriptional RNA folding that would allow individual transcription parameters important to riboswitch function to be isolated and

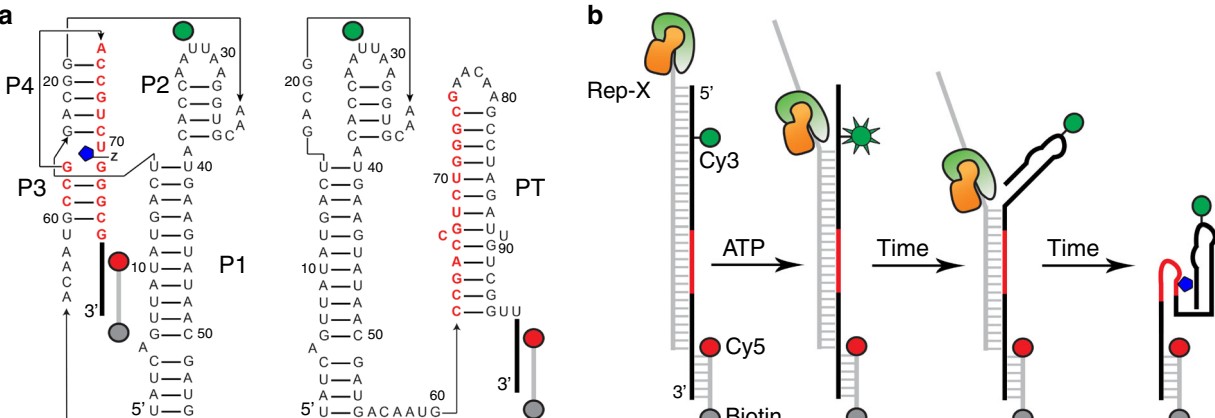

**Fig. 1 ZTP riboswitch single-molecule FRET constructs. a** The ZTP riboswitch aptamer (Δterm, left) formed by paired elements P1-4 bound to ZMP (Z, blue). A Cy3 fluorophore (green sphere) is attached to solvent accessible U32, and a Cy5 fluorophore (red sphere) is placed at the 3′-end of the DNA tether, which is also 5′-end labeled with biotin (gray sphere) for attachment to the streptavidin-coated slide. In the full-length riboswitch (WT, right), the nucleotides colored in red switch between two conformations, one in which a terminator hairpin (PT) or terminator folds and the ligand is not bound. **b** Cartoon depicting the vectorial folding (VF) assay, in which an engineered helicase Rep-X (orange and green) releases the riboswitch RNA (black) in a 5′-to-3′ direction by translocating on the cDNA strand (gray) and unwinding in a 3′-to-5′ direction. During unwinding, when Rep-X is proximal to Cy3, a temporary increase in Cy3 fluorescence [i.e., protein-induced fluorescence enhancement (PIFE)] is observed.

assayed. Here, we use the Rep-X superhelicase[22–24] to mimic sequential (or vectorial) RNA folding without additional polymerase characteristics or limitations in dye placement (Fig. 1b). This allows us to tune the RNA release speed and to introduce pausing at arbitrary positions to test their role in ZTP riboswitch folding without secondary effects arising from transcription termination. As shown below, ZTP riboswitch folding is found to depend on each of these factors.

## Results

**Aptamer domain versus full-length riboswitch folding.** Here, we examined the *F. ulcerans* ZTP riboswitch with and without the termination hairpin (Fig. 1a) using single-molecule and bulk assays. The 75-nt riboswitch aptamer lacking the terminator sequence (Δterm) binds ZMP with an apparent dissociation constant ($K_d$) of ~0.5 μM[25]. However, ZMP binding to the ZTP riboswitch aptamer is sensitive to 3′-end extension after nt 75, and variants containing as few as five base pairs of the terminator bind poorly to ZMP (Supplementary Fig. 1), suggesting that even partial termination hairpin folding competes with ligand binding. Consistent with a recent study on the *Thermosinus carboxydivorans* ZTP riboswitch[26], these measurements suggest that, at equilibrium, there should exist an ~10 nt ligand binding window. We labeled Δterm with donor (Cy3) and acceptor (Cy5) fluorophores[27] and tested for a ZMP-responsive conformational change using single-molecule FRET[28,29] (Fig. 1a), which would be consistent with previous inline probing experiments[6]. When folded using a thermal refolding protocol (Online Methods), Δterm adopted a low-FRET efficiency conformation ($E_{FRET}$ ~0.25) in the absence of ZMP, and shifted to a mid-$E_{FRET}$ conformation (~0.6) upon ZMP binding (Fig. 2a). In the absence of ZMP, Δterm predominantly occupied the low-$E_{FRET}$ conformation but infrequently visited a mid-$E_{FRET}$ conformation like the ZMP-bound state ($E_{FRET}$ ~ 0.6, Fig. 2b), indicating transient conformational changes in the absence of ZMP. The $K_d$ of the fluorophore-labeled Δterm for ZMP (1.3 μM, Fig. 2c) is ~2.5-fold higher than that of the unlabeled Δterm (~0.5 μM)[25], suggesting that fluorophore labeling and surface attachment have only a modest effect on ligand binding. At saturating ZMP (≥1 mM), ~30% of Δterm persisted in the low-$E_{FRET}$ conformation (Fig. 2c), possibly due to a misfolded subpopulation incapable of ZMP binding.

Consistent with equilibrium binding measurements (Supplementary Fig. 1b) in which ZMP is unable to bind to 3′ extended ZTP riboswitch RNAs, the 94-nt wild-type ZTP riboswitch (WT) containing the terminator sequence yielded predominantly a ZMP-unresponsive terminated conformation ($E_{FRET}$ ~ 0.2, Supplementary Fig. 2a) after being thermally refolded in the absence of ZMP. Thermal refolding of the WT construct in the presence of 1 mM ZMP still yielded only ~10% ZMP-bound aptamer-containing conformation ($E_{FRET}$ ~ 0.5, Supplementary Fig. 2a). These data suggest that while some ligand binding can occur, the aptamer-containing conformation cannot be efficiently accessed when the fully synthesized riboswitch is thermally refolded in the absence or presence of ZMP.

To compare with thermal refolding, we folded the aptamer-only construct Δterm and terminator-containing WT construct using a helicase-based vectorial folding assay (Figs. 2d and 3a, Online Methods). The assay mimics cotranscriptional RNA folding by unwinding an RNA/DNA heteroduplex with the extremely processive engineered 3′-to-5′ DNA helicase Rep-X[22], which releases the RNA strand sequentially in the 5′-to-3′ direction, the direction of RNA synthesis during transcription (Fig. 1b). An $E_{FRET}$ ~0 state was observed before initiating unwinding by ATP addition (Figs. 2d and 3a), consistent with

separation of the fluorophores by ~60 base pairs (~20 nm). Incomplete conversion of the duplex is evident after ATP addition as not all duplexes are unwound by Rep-X, and these molecules are excluded from calculation of the aptamer-containing conformation. When the WT RNA was vectorially folded in the presence of ≤0.01 mM ZMP, only the low-$E_{FRET}$ (~0.2) terminated conformation was observed (Fig. 3a). At higher ZMP concentrations, we observed the mid-$E_{FRET}$ (~0.5) aptamer-containing conformation. At 0.1 mM and 1 mM ZMP, 17 and 38%, respectively, of all released RNA showed the aptamer-containing conformation (Fig. 3a), higher than the percentage obtained by refolding of the WT construct (Supplementary Fig. 2a). These results are consistent with bulk single-round transcription termination assays under similar synthesis speeds. In those assays, riboswitch activation during transcription in the presence of 0.2 mM NTPs requires at least 0.2 mM ZMP, a concentration far higher than the $K_d$ of ~0.5 μM for the isolated aptamer at equilibrium (Fig. 3b and Supplementary Fig. 3).

In contrast to the full-length riboswitch, different folding approaches did not affect the folding outcome of the aptamer-alone Δterm construct, as the vectorially folded and thermally refolded Δterm bound ZMP to a similar affinity and extent (Fig. 2c). For the full-length riboswitch, the ZMP-bound aptamer-containing conformation persisted for at least 10 min after unwinding, suggesting that this ligand-bound conformation is stable during our measurements (Supplementary Fig. 2b, c). In the absence of ZMP, transition from the ligand-free aptamer-containing conformation into the terminated conformation must occur rapidly, as addition of 1 mM ZMP 30 s after initiating unwinding did not reveal a ligand-responsive subpopulation (Supplementary Fig. 2d).

**Effects of velocity and pausing on riboswitch folding.** To test the effect of unwinding speed, which can be used to mimic transcription speed, on riboswitch folding, we replaced Rep-X with the helicase PcrA-X, which was previously shown to unwind dsDNA substrates at an ~10-fold slower rate[22]. Compared to Rep-X, PcrA-X yielded a higher fraction of the WT construct adopting the mid-$E_{FRET}$ aptamer folded state, especially at 0.01–0.1 mM ZMP (Fig. 4a, b), suggesting that by giving the aptamer domain more time to fold before the emergence of the termination hairpin, the slower helicase mimics slower transcription. Indeed, in bulk transcription assays, the ZMP concentration required for riboswitch activation ($T_{50}$) also decreased as the apparent rate of transcription decreased at lower NTP concentrations (Supplementary Fig. 2e–g). According to these experiments, the ZTP riboswitch could function like a thermodynamically controlled switch (i.e., $T_{50} = K_D$) if NTP concentration is sub-μM, far below physiological levels. Overall, less ZMP is required for riboswitch activation when more time is allowed for ligand binding, consistent with a kinetic control model for ZTP riboswitch function under these conditions. This observation was consistent between bulk and single-molecule measurements despite the substantial fraction of readthrough leakiness (i.e., readthrough at 0 mM ZMP) evident in the bulk assays, especially at faster transcription speeds (Supplementary Fig. 2e).

Similarly, depending on the location and duration of the pause, transcription pausing could provide additional time for aptamer domain folding and ligand binding[18,30]. For example, if cotranscriptional folding of helices P1-2 (nt 1–54) is slow, a pause at nt 54 after these sequences would promote aptamer formation. In cotranscriptional studies using RNAP[15–17], though it is feasible to modulate the overall tendency of RNAP to pause via addition of Nus proteins or RNAP mutations[9,17,18,20], it is

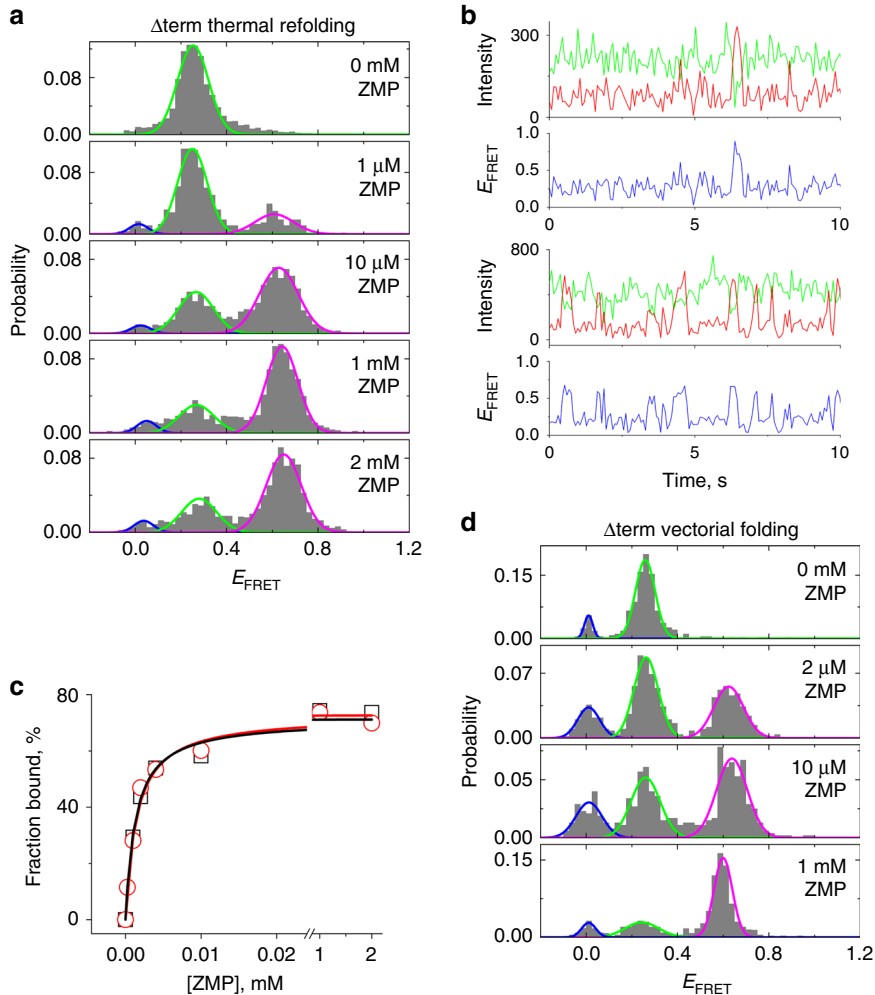

**Fig. 2 ZMP binding to the ZTP riboswitch aptamer after thermal refolding and vectorial folding. a** The 75-nt ZTP riboswitch aptamer domain (Δterm) was thermally refolded at 0 mM ZMP and then incubated with different concentrations of ZMP. Gaussian fitting with global constraints was used to determine the relative population of the ZMP-bound (magenta) vs. ZMP-unbound (green) aptamer. The zero-$E_{FRET}$ state (blue) was also fit to account for a small fraction (~5 %) of annealed RNA due to remaining splint ssDNA from sample preparation. **b** Example single-molecule trajectories of thermally refolded Δterm in the absence of ZMP (top) and in the presence of 1 mM ZMP (bottom). Intensities of Cy3 (green), Cy5 (red), and the resulting $E_{FRET}$ (blue) are shown. **c** The percentage of ZMP-bound aptamer plotted against ZMP concentration and fitted to the Langmuir function for thermally refolded (black) and vectorially folded (red) Δterm. The apparent $K_D$ and maximum percentage ($P_{max}$) were determined to be 1.3 µM and 71%, respectively, for thermally refolded and 1.5 µM and 73% for vectorially folded. **d** $E_{FRET}$ histograms of Δterm after VF at 0 mM ZMP and incubation with different concentrations of ZMP. Here, a multiple turnover VF protocol was used to maximize the amount of unwound Δterm (Online Methods). Global fits were performed as in **b**.

difficult to introduce site-specific pauses at arbitrary positions. In VF experiments, we mimic a specific pause at nt 54 by using a DNA that hybridizes only to nt 55–94 of the WT riboswitch, allowing nt 1–54 to pre-fold (Fig. 4c). When this shorter heteroduplex was vectorially folded, an ~2-fold higher percentage of WT adopted the aptamer-containing conformation under limiting ZMP concentration of 0.1 mM (Fig. 4d). This increase in aptamer-containing conformation was similar to what was achieved by VF with the slower PcrA-X (Fig. 4b, d). In contrast, when we introduced a pause at nt 40, which allows helix P2 but not P1 to equilibrate, the percentage of aptamer-containing conformation remained the same as the full-length heteroduplex (Fig. 4b, e). These results suggest that the slow folding of helix P1 is rate-limiting for aptamer formation. Indeed, sequential folding of the ZTP riboswitch sequence in silico predicts a non-native 6-bp helix to stay folded until nt 54, at which point the 6-bp helix is predicted to dissolve in favor of P1 helix folding (Supplementary Fig. 4a). Recent chemical probing of the *Clostridium beijerinckii*

ZTP riboswitch directly observed the formation of this helix, which occurs in about half of ZTP riboswitches[13]. Natural ZTP riboswitch sequences vary significantly in the sequence between the 3′ end of P1 and the 5′ end of P3 and often contain additional helices that may delay the synthesis of P3-4 and possibly pause RNAP.

Next, we tested how terminator stability affected riboswitch function. In bulk transcription assays, mutation of the first three terminator bases (81–83 A) reduced the ZMP concentration required for riboswitch activation and led to substantially higher readthrough compared to WT in the absence of ZMP (Supplementary Fig. 4b, c). However, it was unclear from the transcription assays whether this mutation affected terminator folding or, if folded, its termination efficiency on RNAP[31,32], because RNAP readthrough depends on both. Using VF experiments, we observed a larger fraction of the mutant riboswitch folding into the ZMP-bound aptamer-containing conformation ($E_{FRET}$ ~ 0.4, Supplementary Fig. 4d, e). The

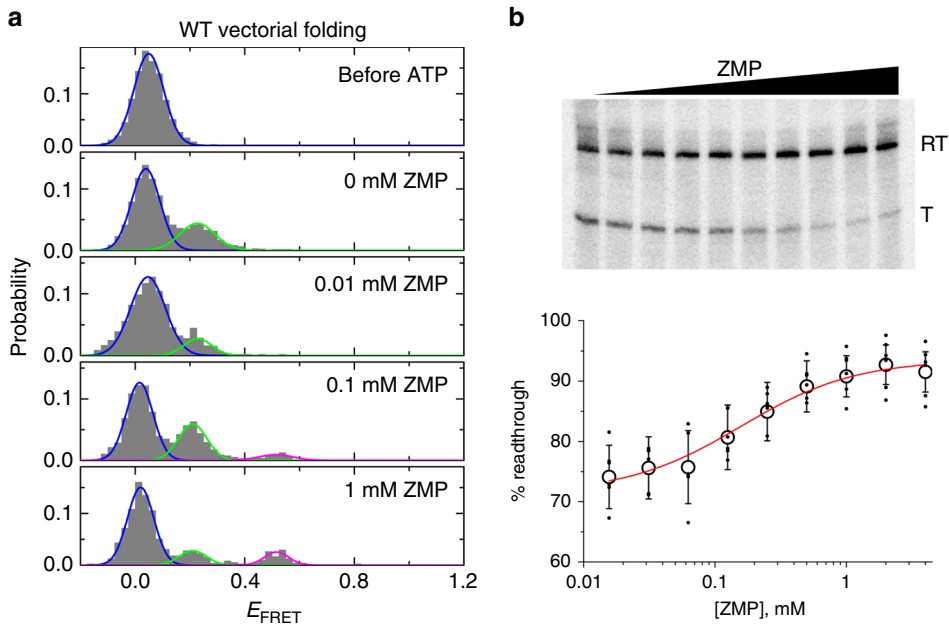

**Fig. 3 Single-molecule helicase-catalyzed unwinding experiments of the ZTP riboswitch. a** The $E_{FRET}$ histograms of the WT heteroduplex before and after ATP addition at different ZMP concentrations using VF with Rep-X. Gaussian fitting with global constraints was used to determine the relative populations of the initial heteroduplex (blue), terminator (green), and ZMP-bound aptamer (magenta) conformations. **b** Bulk single-round transcription termination experiments of ZTP riboswitch are shown in the presence of 0.2 mM NTPs and varying amounts of ZMP (0–4 mM). RT and T stand for the readthrough and terminated products, respectively. Quantification of all experiments under these conditions is shown below (mean ± standard deviation (s.d.) shown as large open circles, $n = 6$ independent experiments shown as small circles).

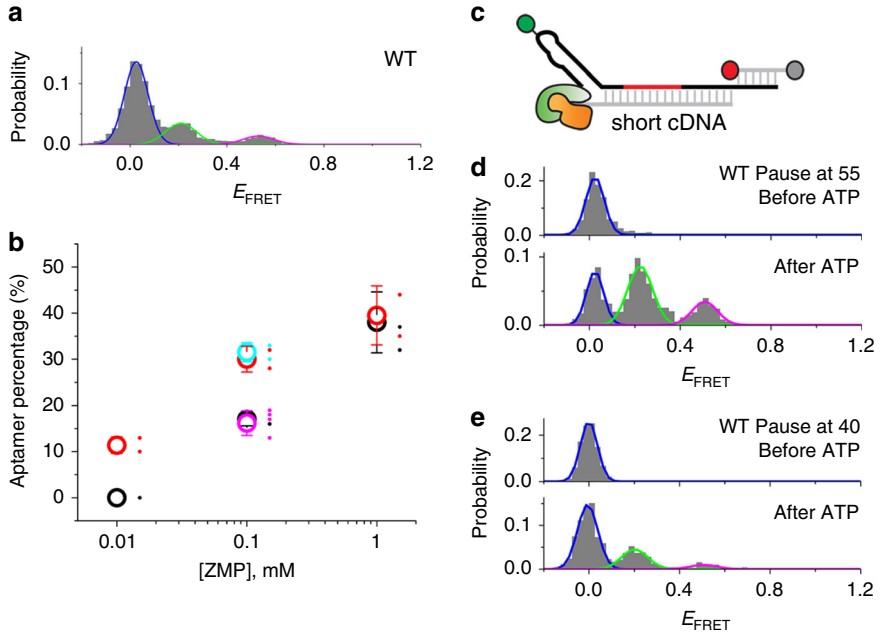

**Fig. 4 The effect of slower unwinding speed and artificial pausing on riboswitch folding. a** VF histograms for WT ZTP riboswitch unwound by PcrA-X in the presence of 0.1 mM ZMP are shown with Gaussian fits to heteroduplex (blue), terminator (green), and aptamer (magenta) populations. **b** The percentage of aptamer-containing conformation obtained by VF with Rep-X (black) and PcrA-X (red) are shown in the presence of 0.01 to 1 mM ZMP. The percentage of aptamer-containing conformation obtained by VF for pause mimics at positions 40 and 55 (magenta and cyan, respectively) are shown. All values are mean ± s.d. shown as large open circles with $n = 2–4$ independent experiments shown as small circles and offset for clarity. **c** Cartoon depicting the pause mimic complex at position 54, in which a cDNA is hybridized only to nt 55–94 of the WT ZTP riboswitch. Addition of ATP initiates unwinding starting from nt 55 in this complex. **d** $E_{FRET}$ histograms of the cDNA:WT heteroduplexes mimicking a pause at position 55 before (top) and after (bottom) addition of ATP in the presence of 0.1 mM ZMP, as measured by VF with Rep-X. **e** $E_{FRET}$ histograms of the pause-mimicking heteroduplexes at position 40 before (top) and after (bottom) addition of ATP in the presence of 0.1 mM ZMP, as measured by VF with Rep-X.

81–83 A mutation did not notably reduce the propensity of fully synthesized RNAs to form the terminator, as 81–83 A still yielded predominantly the terminated conformation upon thermal refolding (~85% vs. ~90% for WT, Supplementary Figs. 2a and 4f). In contrast, mutation of the last three terminator bases (92–94 A) caused more pronounced destabilization of the terminated conformation, as 92–94 A was partially responsive to ZMP at equilibrium after thermal refolding (Supplementary Fig. 4g). Moreover, the double mutant (81–83, 92–94 A) destabilizes the terminator so significantly that the majority of this mutant binds ZMP at equilibrium (Supplementary Fig. 4h). Taken together, disruption of different terminator regions affects folding and switching outcomes both by decreasing stability and possibly by kinetic factors, which is further explored below.

**Single-molecule measurements of unwinding speed**. As the unwinding speed is a critical parameter for relating VF to folding during transcription, and as previous Rep-X rate measurements examined dsDNA templates[22] rather than RNA/DNA hybrid duplexes used herein, we measured helicase speed in three independent ways (Fig. 5a). First, we extended the DNA strand in the heteroduplex to accommodate an additional 18-nt Cy3-labeled oligo (Fig. 5a, top). Upon initiating unwinding, Rep-X first displaced the 18-nt oligo, resulting in ~50% decrease in Cy3 intensity (Fig. 5b). Subsequently, we observed an increase and then a decrease in Cy3 intensity which we attributed to protein-induced fluorescence enhancement (PIFE)[33,34] caused by helicase unwinding through the Cy3 labeling position at U32. PIFE peak centers were determined by Gaussian fitting and used to define the position of Rep-X on the template (Online Methods). The time difference ($\Delta t$) measured between the Cy3 intensity decrease and PIFE peak center corresponds to the time for Rep-X to unwind 32 base pairs of the heteroduplex, from which we estimated the unwinding speed to be $65.0 \pm 0.39$ nt s$^{-1}$ (Fig. 5c).

In a second approach to measure unwinding speed, we positioned two Cy3 fluorophores along the WT riboswitch sequence and excluded the Cy5 fluorophore (Fig. 5a, middle). This labeling strategy resulted in two PIFE peaks during unwinding (Fig. 5d, e), and the $\Delta t$ between the two peak centers, which corresponds to the time for Rep-X to unwind from the first Cy3 position to the second, allowed estimation of unwinding speed using only a single fluorophore. Separation of the dyes by 72 base pairs (Fig. 5d) and 52 base pairs (Fig. 5e) resulted in average $\Delta t$ of $2.01 \pm 0.20$ s and $1.64 \pm 0.15$ s, respectively, and unwinding speeds of $60.4 \pm 4.8$ nt s$^{-1}$ and $52.5 \pm 3.1$ nt s$^{-1}$, respectively.

As the 3′ end of the ZTP riboswitch sequence is more GC-rich than the 5′ end, we used a third method to measure the unwinding speed over different portions of the duplex to assay the effects of GC-richness on unwinding speed (Fig. 5a, bottom). In this method, unwinding releases a Cy5-labeled DNA oligo prior to release of the Cy3-labeled RNA oligo, which is observed by a loss in FRET prior to a loss in Cy3 intensity (Supplementary Fig. 5a, b). The average unwinding speed over different portions of the WT riboswitch sequence was similar (Supplementary Fig. 5a, b), suggesting Rep-X unwinds at a uniform average velocity over the duplex. Importantly, the loss of FRET in this assay is easily distinguished from photobleaching when many traces are examined (Supplementary Fig. 5c), despite appearing identical for any particular single trajectory in this experiment. Also, the average time interval between loss of the first and second oligos in these experiments was unchanged when an RNA oligo was replaced by a DNA oligo and vice versa (Supplementary Fig. 5d, e). Compared to Rep-X, PcrA-X facilitated duplex unwinding more slowly, at a speed of ~8 nt s$^{-1}$ (Supplementary

Fig. 5b), which was also distinguished from photobleaching when many traces were examined (Supplementary Fig. 5f). Thus, the ~6-fold change in unwinding speed between PcrA-X and Rep-X is significant enough to modulate the percentage of aptamer-containing conformation obtained after VF, and the two enzymes can be used to mimic slow and fast RNAP speeds.

**Real-time unwinding measurements to observe folding pathways**. To investigate how folding dynamics results in the riboswitch regulatory landscape observed above, we performed single-molecule experiments to monitor the decision-making trajectory of individual riboswitches in real-time (Online Methods). As in the unwinding speed measurements above, we observed an intensity peak in the Cy3 signal, which we attributed to PIFE[33] upon helicase unwinding through the Cy3 labeling position at U32 (Fig. 6 and Supplementary Fig. 6). By fitting the center of each PIFE peak to a Gaussian, trajectories were synchronized and aligned according to PIFE peak centers, which is critical for comparing trajectories, as unwinding could initiate with different lag times or at different rates after ATP injection. In the presence of 1 mM ZMP, WT and 81–83 A molecules folded into and remained for long periods in the mid-$E_{FRET}$ aptamer-containing conformation (~0.5) although the terminated conformation ($E_{FRET}$ ~0.2) is also observed (Fig. 6a and Supplementary Fig. 6a). In the absence of ZMP, most WT molecules were observed to transition into the low-$E_{FRET}$ (~0.2) terminated conformation within 1.5 s; however, 8% (2/24) of the WT trajectories showed high-$E_{FRET}$ values (≥0.7) for ≥3 s (Fig. 6b), longer than what it takes for Rep-X to complete unwinding given its speed. Using the measured average Rep-X unwinding speed, the progress of Rep-X unwinding through the ZTP riboswitch structural elements (Fig. 1a) vs. time can be estimated (Fig. 6b). Although in the absence of ZMP 81–83 A predominantly formed the terminated conformation 1 min after initiating unwinding (Supplementary Fig. 4d), 60% (9/15) of the 81–83 A trajectories maintained the high-$E_{FRET}$ conformations (≥0.7) for ≥3 s in the initial phase of folding (Fig. 6c), suggesting that formation of the mutant terminator ($E_{FRET}$ ~ 0.2) was severely delayed during VF (also compare $E_{FRET}$ histograms in Fig. 6b, c). Such high-$E_{FRET}$ intermediates (≥0.7), which could functionally promote transcription readthrough if the conformations persisted long enough for RNAP to pass the terminator, provide a plausible mechanism for leaky readthrough (i.e., inefficient termination in the absence of ligand). Taken together, although the 81–83 A mutation only mildly destabilizes the terminator under equilibrium refolding conditions (Supplementary Fig. 4f), it can drastically delay terminator folding, causing readthrough in the kinetically controlled switching landscape.

## Discussion

In order to relate RNA folding during unwinding to folding during transcription, the release rate of RNA during unwinding should be similar to the synthesis rate of RNAP. The three approaches we have used to investigate the unwinding speed (Fig. 5) show that Rep-X releases the RNA at a rate of ~60 nt s$^{-1}$, which is within the 20–80 nt s$^{-1}$ synthesis rate for bacterial RNA polymerases[35] but ~2.5-fold slower than the previously reported rate for Rep-X on dsDNA substrates[22], possibly due to the different stabilities or lengths of the substrates. As the RNA release speed by Rep-X is similar to the speed of RNAP, the vectorial folding assay could be widely adapted to study other kinetically controlled cellular processes such as cotranscriptional assembly of ribonucleoprotein complexes[36]. Likewise, the ~8 nt s$^{-1}$ unwinding rate for PcrA-X suggests that it serves as an analog for slower transcription speeds. Along with the pause mimic

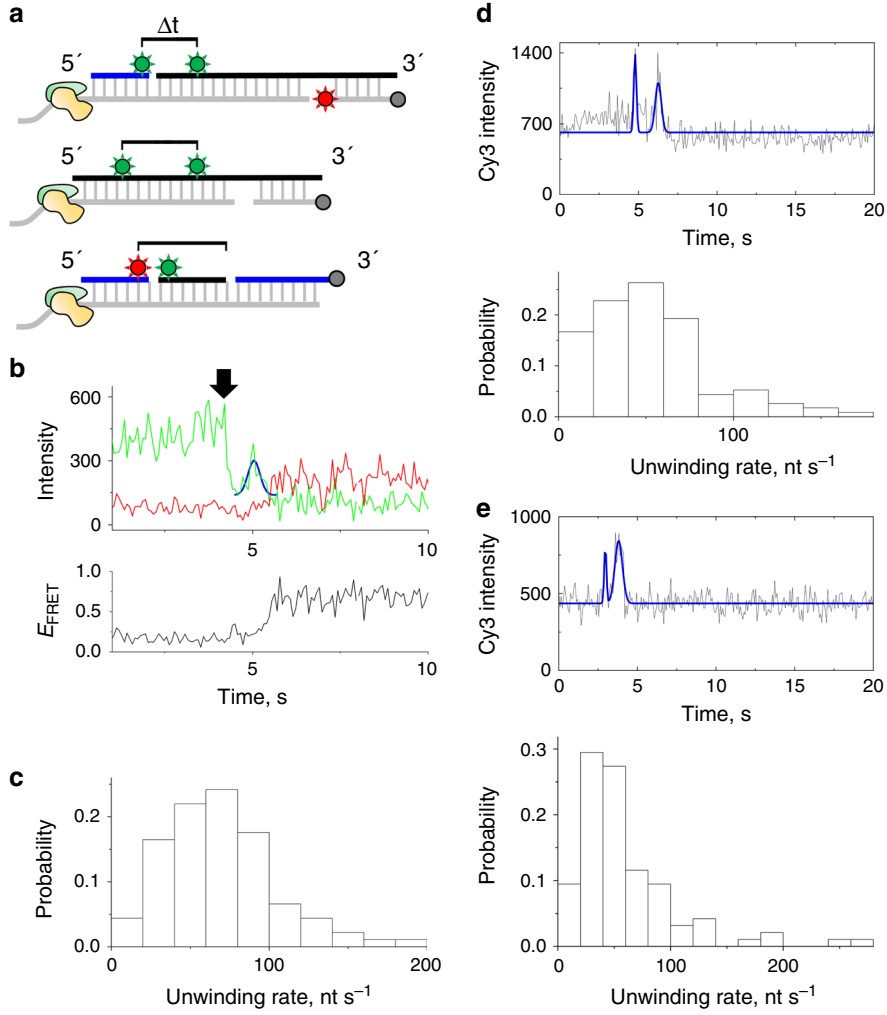

**Fig. 5 Real-time helicase rate measurements. a** Cartoon depicting the three constructs used for rate measurement experiments, either through the loss of Cy3 intensity and PIFE (top), two PIFE peaks (middle), or the loss of FRET and Cy3 intensity (bottom). The cartoon is depicted as in Fig. 1b, with DNA strands in gray, ZTP riboswitch-derived RNA strands in black, other DNA or RNA strands in blue, Cy3 and Cy5 dyes indicated by green and red circles, and biotin labels indicated by gray circles. **b** Individual single-molecule trajectory in the presence of 0.1 mM ZMP for the left scheme, using loss of Cy3 intensity and PIFE as time point indicators. The drop in Cy3 intensity is arrowed, and the PIFE peak fit by a Gaussian curve (blue line). **c** Histogram of helicase rates obtained from all such experiments. The Δt between intensity drop and PIFE peak is shown ($n = 91$ molecules). **d** Individual single-molecule trajectory (top) and histogram of all helicase rates (bottom) for double PIFE experiment with Cy3 labels at U12 and U84 used as time point indicators. For the example trajectory, the raw data (black) and Gaussian fits of PIFE peaks (blue) are shown. The mean time difference (Δt) between PIFE peaks for experiments is $2.01 \pm 0.20$ s (mean ± standard error of the mean (s.e.m.), $n = 95$ molecules). **e** Individual single-molecule trajectory (top) and histogram of all helicase rates (bottom) for double PIFE experiment with Cy3 labels at U32 and U84. The mean Δt between PIFE peaks is $1.64 \pm 0.15$ s (mean ± s.e.m., $n = 112$ molecules).

complexes, these engineered helicases allow for the most complete test of the kinetic control model to date, employing multiple tests of kinetic variations as summarized below.

While the isolated aptamer domain displays reversible binding behavior and could be just as well investigated by equilibrium folding measurements (Fig. 2), the terminator-containing WT construct must be assayed during unwinding to investigate its switching behavior (Fig. 3). Consistent with a kinetic control model, slower unwinding speeds promote ligand-responsive switching as it allows the aptamer more time to fold before the terminator element competes for folding. In cells, if the presence of moderate concentrations of ZMP/ZTP is accompanied by slower transcription speeds, enhanced riboswitch activation should occur in a manner similar to unwinding with the slower helicase PcrA-X. The total pool of ZMP and ZTP ranges from ~0.2 mM to ~1.3 mM in *E. coli* depending on the presence of trimethoprim[4], so even basal levels of Z nucleotides should be

sufficient to activate the ZTP riboswitch if transcription is hindered at the same time. The terminator alone can also be sensitive to transcription speed[20], adding another layer of control if the transcription speed is altered.

VF assays also provided unique insights into the effect of various pause sites. In comparison, conventional RNAP-based methods have difficulties in assessing such effects, because perturbing pause sites requires either mutation of DNA templates, which can have secondary effects on folding (i.e., mutating the terminator hairpin), or addition of proteins that modulate transcription rates (i.e., Nus proteins), which also affect other transcription dynamics. Therefore, it is challenging to probe the effect of specific pause sites in RNAP-based cotranscriptional folding experiments. Using VF assays, we found that using the pause mimics, i.e., initiating unwinding after a slow folding element, promotes riboswitch activation at limiting ZMP concentrations, similar to the effect of slower unwinding speeds. However, using

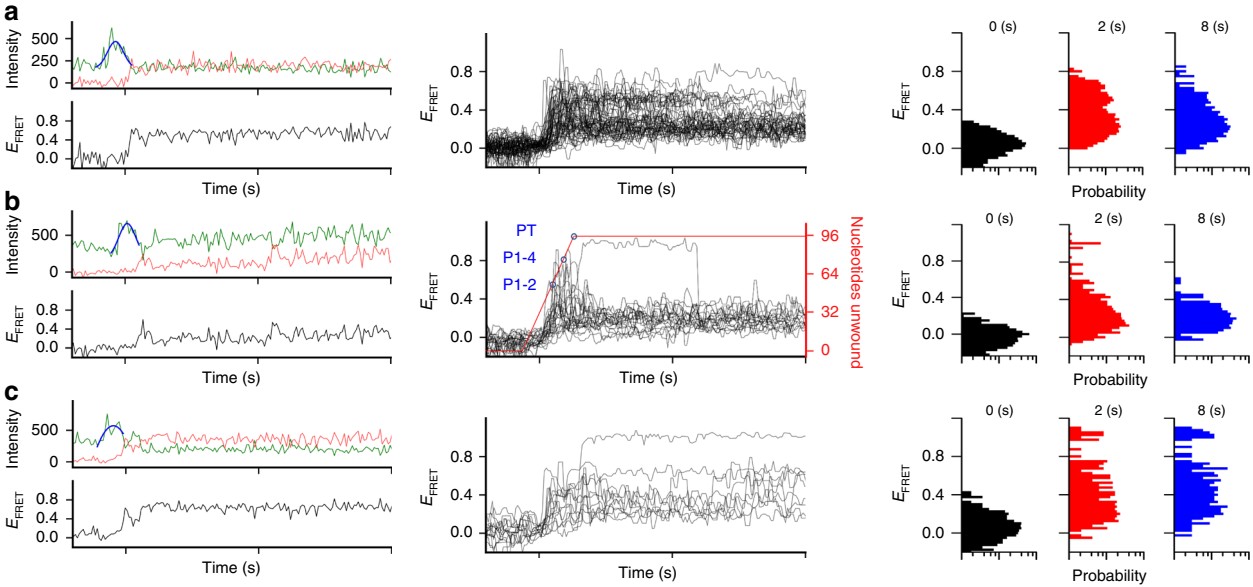

**Fig. 6 Real-time observation of heteroduplex unwinding and riboswitch folding in VF assays. a** Individual real-time trajectory (left), overlay of all WT ZTP riboswitch folding trajectories (center, $n = 56$), and histograms of all trajectories at 0, 2, and 8 s after PIFE (right) using VF with Rep-X in the presence of 1 mM ZMP. The PIFE peaks were fit to a Gaussian (blue) to determine their centers for synchronizing trajectories. **b** Individual real-time trajectory (left), overlay of all WT ZTP riboswitch trajectories (center, $n = 24$), and histograms of all trajectories at 0, 2, and 8 s after PIFE (right) using VF with Rep-X in the absence of ZMP. The relative unwinding position is indicated by a red line, based on the experimentally determined Rep-X unwinding rate and the location of U32 on the template. Blue circles indicate time points corresponding to the unwinding and release of helices P1 and P2, helices P1-P4, and terminator helix PT. **c** Real-time trajectories of terminator mutant 81-83 A in the absence of ZMP using VF with Rep-X, showing an individual trajectory (left), all trajectories overlaid (center, $n = 15$), and histograms of all trajectories at 0, 2, and 8 s after PIFE (right). The time duration between ticks is 5 s. Histogram probabilities are on a log scale, with ticks of 0.1, 1, and 10 percent.

complexes similar to the pause mimics here (Fig. 4a), pauses close to the 3′ end of the riboswitch are difficult to study due to the low melting temperature of the DNA strand that must anneal to the riboswitch RNA. One solution would be to extend the 3′ end of the riboswitch sequence and move the Cy5 labeling position accordingly.

From these experiments, an individual riboswitch is observed to fold through one of many distinct channels or pathways, and the entire folding landscape is made evident from many observations (Fig. 6). This includes the well-populated ligand-bound and ligand-free states, as well as less common pathways. While delayed termination hairpin folding events are infrequent for the WT ZTP riboswitch, weakening the terminator hairpin (81–83 A) increases the observation of these events. Those mutations could cause delayed termination hairpin folding through a mechanism that delays strand displacement of the aptamer by the terminator by three base pairs, assuming the proposed model[13] in which strand displacement occurs in the direction of transcription. That is, rather than initiating at G81, which is the most 5′ nucleotide of the terminator, terminator folding via strand displacement cannot begin until U84. As the terminated conformation is still the predominant state in equilibrium measurements of the 81–83 A variant, the kinetic delay, rather than the stability perturbation, is a more important factor for increasing the readthrough observed in the bulk transcription assays. The findings here highlight the advantages of single-molecule approaches, in which folding heterogeneity can be directly observed in real time. VF assays circumvented the challenge of attaching fluorophore on nascent RNA chains encountered in RNAP-based single-molecule FRET (smFRET) cotranscriptional riboswitch folding studies, and thus making it possible to observe a much wider range of folding dynamics in real time. Future in vivo structural mapping studies can further validate the existence and roles of these intermediates.

A potential drawback of VF assays is that replacing RNAP with a helicase cannot recapitulate some RNAP-specific interactions with nascent chains, though it helps to more clearly dissect the contributions of other key kinetic parameters. In addition, we cannot rule out the effect of potential contacts between the unwound ssDNA and RNA chains, though comparisons made with bulk kinetics and steady-state single-molecule kinetics here and elsewhere[23] suggest that such an effect is not significant at least in the systems we have examined so far. These interactions between ssDNA and RNA chains can also occur with RNAP, such as those in R-loop structures[37]. Thirdly, the current unwinding yield of VF assays is ~30%, which is primarily due to the fact that not every construct has a helicase bound at the helicase concentrations we used, and some helicase dissociates before it begins unwinding[23]. Although 30% should be sufficient for many studies, we anticipate that future developments will improve the yield.

In both bulk transcription and single-molecule unwinding assays, only ~40% of this riboswitch is modulated by ligand binding, suggesting that a large fraction of RNA falls into leaky terminated or leaky readthrough states that do not respond to ligand. As shown here, folding intermediates that prevent proper folding of the aptamer domain promote leaky termination (Fig. 2), and delayed folding of the termination hairpin can bias the riboswitch and promote leaky readthrough (Fig. 6). While ZTP riboswitch folding intermediates may be less conserved than the canonical ligand-bound and ligand-free states[13], targeting them with small molecules is a strategy that would, for example, promote riboswitch termination through stabilizing an off-pathway intermediate that competes with aptamer domain folding. This approach is analogous to targeting short-lived excited states rarely accessed by a dynamic RNA[38,39], although riboswitch transcription intermediates are non-equilibrium states that can temporarily dominate a folding pathway. In addition to lower

conservation, a drawback in targeting riboswitch transcription intermediates with synthetic small molecules is that the new ligand cannot be developed from the canonical riboswitch ligand and thus the target binding site in the intermediate may have lower information content[11] than the canonical binding site.

As kinetically controlled switching behavior is observed for this riboswitch with two different motile enzymes (Rep-X and RNAP), the RNA sequence itself contains the kinetic control information, which is interpreted or manifested by the motile enzyme. Consequently, a conserved riboswitch sequence could still function kinetically in the context of different cellular processes (i.e., transcription, translation), as long as a motile enzyme is present to impose sequential movement. Interestingly, certain translational riboswitches exhibit kinetically controlled folding behaviors[16]. One implication is that cellular RNA helicases or even ribosomes can recommit the fate of these translational riboswitches long after transcription has completed. As many nucleic acid folding processes are kinetically controlled, including those involving ribonucleoprotein complexes[40,41], we believe this method is broadly applicable to many types of RNA and even DNA structures, as it has been demonstrated in ribozymes (e.g., Twister[23]), RNA aptamers (e.g., Spinach and Mango[24]) and even telomeric DNA sequences in an experimental format that mimics co-reverse transcriptional telomeric DNA folding[36], and to the studies of nucleic acid-protein complex assemblies.

## Methods

**Sample preparation for single-molecule measurements**. The following method was used to prepare fluorescently labeled RNA and RNA-DNA complexes[27]. Briefly, all steps involving fluorescently labeled oligomers were performed in the dark. Unlabeled DNA oligomers, primers, and plasmids (Supplementary Table 1) were purchased from Eurofins Genomics LLC and Integrated DNA Technologies. Fluorescently labeled oligonucleotides were purchased from Trilink (San Diego, CA), including the Cy3-labeled 5′-end segment of the *F. ulcerans* ZTP riboswitch (nt 1–33, Supplementary Table 1) and the Cy5-labeled and biotinylated tether DNA (biotin-DNA-Cy5, Supplementary Table 1). Fluorophores were attached to these oligomers via 6-carbon amino linkers at the C5 position of U32 and at the 3′-end of the biotinylated DNA, respectively. ZTP riboswitch RNAs lacking the first 33 nucleotides (Δ1–33) were transcribed and purified from DNA templates PCR amplified from plasmids[25,42]. Mutations to the terminator stem were made via PCR amplification using a reverse primer containing the mutations. Δ1–33 RNAs were phosphorylated with T4 polynucleotide kinase (New England Biolabs), phenol-chloroform extracted, and ethanol precipitated. The Cy3-labeled nt 1–33 RNA and Δ1–33 RNAs were annealed to a DNA splint identical to the cDNA-dT$_{20}$ used in single-molecule measurements and ligated using T4 RNA ligase 2 (New England Biolabs). Ligated RNAs were purified by 8% denaturing urea-PAGE, recovered via elution in Whatman Elutrap electroelution systems, concentrated, washed with DEPC-treater water, and filtered in 0.22 μm spin filters. Concentrations were calculated by the absorbance of Cy3 at 552 nm ($\varepsilon_{552} = 0.15\,\mu M^{-1}\,cm^{-1}$) and the absorbance of the RNAs at 260 nm ($\varepsilon_{260} = 1.16\,\mu M^{-1}\,cm^{-1}$ for the terminator-containing RNAs and $\varepsilon_{260} = 0.982\,\mu M^{-1}\,cm^{-1}$ for the aptamer-only RNA) after correcting for the contribution of Cy3 absorbance at 260 nm (~5% correction). Ligated RNAs were aliquoted and stored at −20 °C in the dark until further use.

Fluorescently labeled RNAs for double PIFE helicase rate measurements (Fig. 5d, e) were constructed in a similar manner, resulting in RNAs identical to the WT sequence and containing two Cy3 labels. For these experiments, three RNAs were ligated, consisting of segments spanning riboswitch nucleotides 1–33, 34–71, and 72–94. The segment 1–33 contained a Cy3 label either at position U12 or position U32. The segment 72-94 contained a Cy3 label at position U84 and also contained the RNA sequence complementary to the biotinylated DNA.

To form annealed heteroduplexes for single-molecule measurements, 20 μl reaction aliquots containing a 1:0.8:9:10 ratio of Cy3-RNA:biotin-DNA-Cy5:cDNA-dT$_{20}$ (160 nM Cy3-RNA, 128 nM biotin-DNA-Cy5, and 1.6 μM cDNA-dT$_{20}$) were prepared in the buffer containing 50 mM HEPES-KOH (pH 7.4) and 150 mM KCl. The reaction aliquots were slowly cooled from 95 °C to 20 °C in a thermocycler (30 s per 5 °C). Annealing success was evaluated on 8% nondenaturing PAGE (in 1X TBE) by comparison to reactions lacking either biotin-DNA-Cy5, cDNA-dT$_{20}$, or both. Reactions lacking cDNA-dT$_{20}$ were also saved for single-molecule measurements. Gels were scanned with a Typhoon Trio Variable Mode Imager (GE Healthcare) to identify bands containing fluorescent labels.

For dropoff-PIFE experiments (Fig. 5b, c), a longer cDNA-dT$_{20}$ (cDNA-speedmer-dT$_{20}$) was used for annealing reactions in the presence of an 18-nt RNA oligo labeled at the 3′ end with Cy3 (18-speedmer), the WT Cy3-labeled RNA, and biotin-DNA-Cy5. Reactions contained a 1:0.8:9:10 ratio of

Cy3-RNA:biotin-DNA-Cy5:cDNA-speedmer-dT$_{20}$:18-speedmer to saturate cDNA-speedmer-dT$_{20}$ with 18-speedmer. Annealing success was evaluated via nondenaturing PAGE as above.

For dropoff experiments (Supplementary Fig. 5), 20 μl reaction aliquots containing a 1:1.2:1.2:1.2 ratio of biotin-DNA:cDNA-dT$_{20}$:Cy3-oligo:Cy5-oligo (1 μM biotin-DNA, 1.2 μM cDNA-dT$_{20}$, 1.2 μM Cy3-oligo and 1.2 μM Cy5-oligo) were prepared in the buffer containing 50 mM HEPES-KOH (pH 7.4) and 150 mM KCl. The reaction aliquots were slowly cooled from 95 °C to 20 °C in a thermocycler (30 s per 5 °C). Selecting doubly labeled surface-tethered molecules ensured that the molecules being imaged contained all four oligos.

**Single-molecule measurements**. Vectorial folding (VF) was performed with the following method[23]. Briefly, Rep-X was incubated in the loading buffer for 5 min with heteroduplexes that were immobilized on the imaging surface. The loading buffer contained 50 nM Rep-X, 50 mM HEPES-KOH (pH 7.4), 150 mM KCl and 10 mM MgCl$_2$. Free Rep-X was washed out and unwinding was initiated by adding the unwinding buffer. The unwinding buffer contained 50 mM HEPES (pH 7.4), 150 mM KCl, 10 mM MgCl$_2$, 1 mM ATP, and different concentrations of ZMP. To stabilize the remaining aptamer conformation and distinguish it from the terminator conformation, 1 mM ZMP was supplied into the imaging channel after 1 min of vectorial folding. To observe the heteroduplex unwinding and riboswitch folding in real-time, imaging was started ~5 s before the addition of the unwinding buffer. The loading and unwinding buffers used during imaging contained additional 4 mM Trolox, 0.8 % wt vol$^{-1}$ glucose, 165 U ml$^{-1}$ glucose oxidase, and 2170 U ml$^{-1}$ catalase to reduce the photobleaching rate[43].

For VF multiple turnover experiments with Rep-X (Fig. 2d), 50 nM Rep-X was included in the unwinding buffer and no separate Rep-X loading step was performed. When PcrA-X was used in place of Rep-X, 100 nM of PcrA-X was added in the loading buffer. Other buffer components were kept the same.

To refold RNAs for single-molecule measurements, 20 μl reaction aliquots containing a 1:0.8 ratio of Cy3-RNA:biotin-DNA-Cy5 (160 nM Cy3-RNA and 128 nM biotin-DNA-Cy5) were prepared in the buffer containing 50 mM HEPES-KOH (pH 7.4), 150 mM KCl, 10 mM MgCl$_2$ and different concentrations of ZMP. The reaction aliquots were slowly cooled from 95 °C to 20 °C in a thermocycler (30 s per 5 °C). When refolded at 1 mM ZMP, RNAs were imaged within 5 min after refolding to minimize the loss of ZMP-bound aptamer due to terminator hairpin formation.

**Single-molecule data acquisition and analysis**. Single-molecule data acquisition and analysis were performed with the following method[23,44]. Briefly, fluorescently labeled samples were immobilized on DT20 or PEG surfaces through a biotin-NeutrAvidin linkage[45,46]. Samples were imaged with a prism-based total internal reflection fluorescence (TIRF) imaging microscope[47]. A 532-nm laser (Coherent Compass 315 M) and a 633-nm laser (Research Electro-Optics) were used for Cy3 and Cy5 excitation, respectively. A water immersion objective (NA 1.2, 60X, Olympus) and an EMCCD camera (Andor Technology IXon 897) were used to collect and record signals with 0.075 s time resolution. The fluorescence emission was filtered by a long pass filter (Semrock BLP02-561R-25) and a notch filter (Chroma ZET635NF) to block the excitation lasers. Spatially separated single molecules were picked by a custom IDL code and their intensities were extracted for further analysis. To plot the trajectory overlay, we synchronized trajectories according to the PIFE peak center. The percentage of aptamer-like fold is calculated as the ratio of the aptamer population (magenta) over the sum of the terminator (green) and aptamer populations.

For traces containing multiple PIFE peaks (Fig. 5d, e), peak centers were determined by Gaussian fitting in Origin (OriginLab, Northampton, MA), and time differences between PIFEs (Δt) were calculated by the difference between the peak centers. Traces appearing to contain too many PIFE peak were excluded from analysis (7 traces excluded from 12–84 dataset, 8 traces from 32–84 dataset, and 9 traces from drop-PIFE dataset). Unwinding rates are reported as the mean ± s.e.m. with the *n* being the number of individual molecules.

For dropoff-PIFE traces (Fig. 5b, c), the drop in Cy3 intensity was determined by eye, and time was recorded. PIFE peak centers were determined by Gaussian fitting as above. The time difference (Δt) between these two events was calculated for each trace and used to determine a helicase unwining rate (in nt s$^{-1}$) with the known distance of 34 nt separating the two events. Traces containing more than one PIFE peak were excluded from analysis (9 traces excluded). The average rate is reported as the mean ± s.e.m. with *n* being the number of individual molecules (91 total).

For dropoff experiments (Supplementary Fig. 5), the decrease in FRET value and decrease in Cy3 intensity were determined by eye. The time difference (Δt) between these two events was calculated for each trace, and the average unwinding rate was calculated using all traces for which Δt was less than 2 s for Rep-X (or 10 s for PcrA-X), which includes traces contained within the Gaussian distribution of rates.

**Single-round transcription termination assays**. PCR templates were amplified from a plasmid containing a 100 nt leader sequence, a λ Pr promoter, 26-nt C-less region, the *F. ulcerans* ZTP riboswitch sequence, which contains the aptamer

domain, expression platform and 51 nucleotides beyond the terminator U8 sequence. After amplification, templates were purified by 2% agarose gel electrophoresis. Mutations to the templates were made by site-directed mutagenesis to the plasmid and confirmed by sequencing.

Transcription reactions were performed by halting transcription by omission of CTP and restarting transcription by adding NTPs and ZMP[25]. Halted transcription complexes contained 80 pmol uL$^{-1}$ DNA template, 20 mM Tris-HCl, pH 8, 20 mM NaCl, 4 mM MgCl$_2$, 0.1 mM DTT, 0.1 mM EDTA, 4% glycerol, 0.14 mM ApU, 1 μM GTP, 2.5 μM ATP, 2.5 μM UTP, ~1 μCi mL$^{-1}$ [α-$^{32}$P]-ATP, 0.04 U uL$^{-1}$ *Escherichia coli* RNA polymerase holoenzyme (Epicenter), and 10 μM of a 26 nt oligonucleotide complementary to the C-less region. Transcriptions were restarted by addition of various concentrations of NTPs and ZMP prior to incubation at 37 °C for the duration of the time course, removing aliquots as needed and quenching reactions by addition of loading dye containing 8 M Urea, 20% sucrose, 0.1% SDS, 0.01% bromophenol blue, 0.01% xylene cyanol and placing them on ice or storing them at −20 °C. Reactions were then separated via 8% denaturing urea-PAGE and analyzed[48,49]. To determine apparent reaction rates for appearance of terminated and readthrough transcription products, time courses were fit by single exponential fits using Origin. For single time point titrations, reactions containing different concentrations of ZMP were incubated for 20 min at 37 °C, and results were fit to a 1:1 binding isotherm to determine $T_{50}$ values.

Sequential folding in silico was performed by using Mfold[50] with different lengths of ZTP riboswitch transcripts as inputs.

**Isothermal titration calorimetry (ITC)**. All ITC measurements were performed and analyzed with the following method[51], using a MicroCal iTC200 (Malvern, Egham, UK). Briefly, RNAs were refolded by heating at 95 °C for 2 min and immediately placed on ice. MgCl$_2$ was added to 10 mM, the sample was brought up to volume, and the RNA was incubated at 37 °C prior to performing titrations at 37 °C in a buffer containing 50 mM Hepes-KOH, pH 7.4, 150 mM KCl, and 10 mM MgCl$_2$. Typically the cell containing 20 μM RNA was titrated with 200 μM ZMP, but higher concentrations were used for weaker binders.

**Reporting summary**. Further information on research design is available in the Nature Research Reporting Summary linked to this article.

## Data availability
All data are available from the corresponding authors upon reasonable request. Source data are provided with this paper.

## Code availability
The software used for data acquisition is available at http://ha.med.jhmi.edu/resources/ or upon request. The software used for data analysis is custom made and is available upon request.

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

## Acknowledgements

We thank Drs. G. Piszczek and D. Wu in the National Heart, Lung and Blood Institute (NHLBI) Biophysics Core for ITC support. This work was supported by U.S. NSF grant PHY1430124 and NIH grant R35 GM122569 to T.H. C.P.J. is the recipient of NIH grant K22 HL139920. T.H. is an investigator of the Howard Hughes Medical Institute. This work was supported in part by the intramural program of the NHLBI, NIH.

## Author contributions

B.H., C.P.J., A.R.F., and T.H. designed the project, B.H., J.M., P.J.M., and R.R. performed the single-molecule experiments, C.P.J. prepared single-molecule constructs and performed bulk experiments, B.H. and C.P.J. analyzed the data, and B.H., C.P.J., A.R.F., and T.H. wrote the manuscript.

## Competing Interests

The authors declare no competing interests.
