## [Peer Review File · Nature Communications]

REVIEWER COMMENTS

Reviewer #1 (Remarks to the Author):

Hua et al. applied a single-molecule vectorial folding assay to monitor the ZTP riboswitch folding kinetics in real-time. By using the engineered superhelicase Rep-X, they mimicked the passway of RNA transcription and observed that ZTP riboswitch is kinetically controlled with dynamic changes during RNA releasing. The method that they developed is very interesting, because they can modulate the “transcription” speed, the pausing position, and test the precise RNA sequence without affecting the interaction with RNAP. Their approaches may provide a new antibiotic discovery that targets the intermediates of riboswitches. Although the work is very important and the experiments are well designed, I believe the following comments may improve the significance of the study for broader audience.

Major points

1) A major drawback of this manuscript is that it is unclear what is the new insight of this study. It is important to describe the new biological or biochemical insights about ZTP riboswitches. For example, Is this the first report that finds many intermediates of riboswitches including RNA release speed, ZTP riboswitch folding without secondary effects from transcription termination? And can only this approach address to reveal the intermediate states they found? Compared with previous studies, why this method should be applied ? And do these intermediates that the authors observed truly affect the riboswitch conformation and biological function in vivo?

2) Related with the comment 1), do the folding dynamics revealed in this study reflect in vivo situation? If so, similar intermediates may also be found by in vivo SHAPE-map or related approaches that probe RNA structures. In other words, RepX-based RNA unwinding approach is very interesting, but it is unclear whether this system really reflects the situation of naturally occurring riboswitches or not.

3) The generality of this approach is unclear. The authors verified the fluorophore labeling did not affect ligand-RNA binding so much. But it is unclear whether each RNA sequences used in this study show similar tendencies. Also, the difference between the helicase and RNAP may affect the parameters of RNA kinetics, transcription speed, and RNA folding and ligand binding. Can this unwinding method be applied to investigate other types of riboswitches? To show the generality of the method, the authors should test their approaches to other riboswitches (or RNA structures).

Minor points

1) In Figure1b legend, the abbreviation VF is not explained.

2) In Figure2a, the blue line should be explained.

3) In Figure3b, the number above the gel image is not described in legend. The number of plot in quantification is 9, but the number of gel image is 10.

4) 4. In figure6, the attachment of RNA model (structure) images is helpful for understanding the kinetic

changes of RNA folding.

Reviewer #2 (Remarks to the Author):

In this manuscript, Hua and Jones, et al. investigate the co-transcriptional RNA folding mechanism of the *F. ulcerans* ZTP riboswitch at single-molecule resolution using a vectorial RNA folding assay that was previously developed by the Ha lab. The vectorial folding assay uses a processive helicase (Rep-X) to sequentially release a fluorophore-labeled RNA from an RNA/DNA heteroduplex thereby mimicking co-transcriptional RNA folding. This approach enables vectorial RNA folding kinetics to be quantified by single-molecule FRET, which is not easily achieved using RNA polymerase (RNAP)-based assays. This work presents the first application of vectorial RNA folding to study riboswitch RNA folding kinetics.

Under equilibrium conditions, the authors observe that the ZTP aptamer domain folds into distinct conformations in the absence and presence of ZMP. These conformations correspond to ZMP-unbound and ZMP-bound states. If the transcription terminator domain is present, the ZMP-bound aptamer-folded state can only be accessed by vectorial folding. This suggests that *F. ulcerans* ZTP riboswitch folding is kinetically controlled and is consistent with the authors' measurements from corresponding bulk in vitro transcription assays. The authors then perform a series of experiments showing that unwinding velocity (which simulates transcription speed), pre-equilibration of defined RNA segments (which simulates transcription pausing), and terminator hairpin stability all contribute to RNA folding outcomes. Together these data strongly support a kinetic folding mechanism. Lastly, the authors measure ZTP riboswitch folding in real-time and are able to quantify differences in terminator folding rates between the WT riboswitch and a mutant with a disrupted terminator stem. Notably, the 81-83A mutant only destabilizes terminator folding under vectorial conditions. This suggests that the deficiency observed for the 81-83A terminator in bulk transcription assays is the result of a kinetic RNA folding defect, rather than terminator hairpin destabilization.

The authors' experiments are convincing, their conclusions are well-supported, and the findings are presented clearly. Both the application of the vectorial RNA folding assay to study a riboswitch RNA and the authors mechanistic findings are significant advances and will be of broad interest. Experimentally, this work is suitable for publication in Nature Communications as-is. However, the authors should consider the points below, which may improve the manuscript.

Major comments:

My only significant suggestion is that the authors present a fairer comparison of their vectorial folding assay to alternative measurement approaches in the introduction. The authors accurately identify the limitations of bulk approaches (structure probing, NMR, and fluorescently labeled ligands) and alternative single-molecule approaches (optical tweezers) for measuring riboswitch RNA folding; the manuscript would be improved if in addition to describing the advantages of their own methodology

they also acknowledge its limitations (which, to be clear, do not impact the validity of the presented work).

While I agree that simulating co-transcriptional RNA folding by helicase unwinding has some distinct advantages relative to RNAP-based assays, the absence of polymerase characteristics can also be viewed as a limitation because it is well-established that interactions between nascent RNA and a transcribing RNAP have quantifiable effects on RNA folding. Additionally, the suggestion that the vectorial folding assay is less susceptible to RNA mutations than RNAP-based assays (lines 83-87) does not seem correct because the labelled RNA is assembled from a transcript synthesized by an RNAP (which should have the same error rate as RNAP-based assays) and a chemically synthesized RNA oligonucleotide (which likely has a worse error rate than typically used RNAPs); unless the authors have some other justification for this statement it should probably be removed.

Lastly, how does the emergence of ssDNA from Rep-X as the RNA/DNA heteroduplex is unwound impact RNA folding? Presumably the ssDNA can interact with the ssRNA at some frequency given that these nucleic acids are complementary and in close proximity. If this is a relevant consideration, the authors should state how it is addressed.

Minor Comments:

1. In Figure 1B it would be helpful to have labels indicating the ZMP condition on the plots.
2. In Figure 3B, what do the numbers above the gel lanes indicate? Unless these numbers have a purpose, it might be better if the authors use a black triangle and indicate ZMP concentration is increasing from left to right.
3. On line 296, should 'absence of trimethoprim' be 'presence of trimethoprim'?
4. In Supplementary Figure 2E the authors should consider showing the full y-axis range on both plots so that readers can directly compare the 10 and 200 micromolar NTP conditions. This is important since comparison of these results supports the authors finding of a kinetic folding mechanism.
5. In Supplementary Figure 3, it would be helpful if panels E, G, and H had better annotations.

Reviewer #3 (Remarks to the Author):

In this paper, Hua et al. used a single-molecule vectorial folding assay to mimic the cotranscriptional folding of a riboswitch, taking advantage of an engineered superhelicase developed by the same group that sequentially releases single-stranded RNA from a heteroduplex at a speed comparable to in vivo transcription. This assay allowed them to monitor the riboswitch folding behavior under different

kinetically controlled conditions, including unwinding speed, position-specific pausing, and terminator strength. The results suggest that a longer time window before terminator formation promotes ligand binding to the riboswitch, leading to its activation.

This elegant work overcomes several limitations of previous studies of cotranscriptional folding, contributes important insights into the kinetic control of riboswitch folding, and can be applied to many other folding systems. The work involves a variety of cleverly designed assays and constructs, showcasing the strength of single-molecule fluorescence spectroscopy in dissecting dynamic biological processes. This paper will be of general interest to researchers in the fields of single-molecule biophysics and gene regulation.

The main concern on the paper is regarding its data presentation. Although the introduction and discussion are nicely written, the results and figures are currently presented in a way that is difficult to follow. Some interpretations also need to be clarified. These issues should be addressed before publication.

Specific comments:

- (1) The study uses many different constructs. They should be clearly depicted in the figure panels (e.g., Fig 2A, 3A, 4A, 4D, 4E, S2, S3D-S3H).
- (2) The zero FRET state in Fig 2D and 3A was interpreted as the starting substrate conformation for the vectorial folding assay. Why is its occupancy much higher for the WT substrate than the Δ_{term} substrate? What is the zero FRET state in the thermal folding assay in Fig 2A? Are those donor-only molecules? If so, how were they distinguished from the conformationally extended molecules?
- (3) Judging from Fig 3A, a large fraction of molecules did not respond to helicase unwinding. What is the explanation for such a low efficiency?
- (4) Are the green (0.2 FRET) and magenta (0.5 FRET) peaks in Fig 3 considered to be in the same structural configurations as the green (0.25 FRET) and magenta (0.6 FRET) peaks in Fig 2? If not, they should be shown in different colors.
- (5) Sample FRET trajectories should be added to Fig 2D, 3 and 4, as they are important for differentiating static heterogeneity from dynamic heterogeneity among FRET states. For example, in Fig 3A, are there interconversions between the 0.2 and 0.5 FRET states for the same molecule? This would provide support to the statement that “this ligand-bound conformation is stable during our measurements” (Line 150).
- (6) I appreciate the multiple ways to calculate the unwinding rate in Fig 5. However, they largely arrived at the same results. I am not sure if they all need to be in the main figure. Furthermore, the distributions for the unwinding rate in Fig 5 are all quite broad, suggesting substantial molecule-to-molecule heterogeneity. This should be taken into consideration when calculating the unwinding time later in the manuscript. For example, the fast and slow molecules may take different folding paths in the real-time experiments in Fig 6.
- (7) The real-time unwinding experiment in Fig 6 is the most exciting part of this study. However, the data are not sufficiently analyzed. How many FRET states were observed in the trajectories? Judging from the overlaid traces, there seem to be at least three states (~ 0.2 , ~ 0.5 , and ~ 0.8). What do they each

correspond to? How do their lifetimes differ between the WT and 81-83A mutant? In the abstract the authors mentioned “an intermediate responsible for delayed terminator formation.” Which state does it refer to? Ideally, the results should be presented in alternative folding pathways, each with a representative trajectory and illustration. While I understand that the statistics are low, which may prevent the authors from making more quantitative conclusions, a more in-depth analysis would still help to better understand the complexity of the system. At a minimum, the “delayed formation of the mutant terminator” should be supported by a more rigorous quantitation.

(8) A model figure summarizing the folding behaviors under different kinetic conditions would be very helpful. Can the different FRET states be related to specific predicted structures of the ZTP riboswitch?

(9) There are a number of mislabels and omissions, especially in the supplementary figures. To name a few:

Fig S2C: circles were not labeled.

Fig S3D: should be 0 mM and 0.1 mM ZMP rather than ATP.

Fig S3E: what do the blue and black circles stand for?

Fig S3 legend: the 87C variant was not described anywhere in the text or figures.

Fig S4A: it was not explained how the traces were segmented to calculate the unwinding times.

We thank the reviewers for their insightful comments and criticism. We have addressed all comments point-by-point below (in red), citing changes to the manuscript where relevant.

REVIEWER COMMENTS

Reviewer #1 (Remarks to the Author):

Hua et al. applied a single-molecule vectorial folding assay to monitor the ZTP riboswitch folding kinetics in real-time. By using the engineered superhelicase Rep-X, they mimicked the pathway of RNA transcription and observed that ZTP riboswitch is kinetically controlled with dynamic changes during RNA releasing. The method that they developed is very interesting, because they can modulate the “transcription” speed, the pausing position, and test the precise RNA sequence without affecting the interaction with RNAP. Their approaches may provide a new antibiotic discovery that targets the intermediates of riboswitches. Although the work is very important and the experiments are well designed, I believe the following comments may improve the significance of the study for broader audience.

Major points

1) A major drawback of this manuscript is that it is unclear what is the new insight of this study. It is important to describe the new biological or biochemical insights about ZTP riboswitches. For example, is this the first report that finds many intermediates of riboswitches including RNA release speed, ZTP riboswitch folding without secondary effects from transcription termination? And can only this approach address to reveal the intermediate states they found? Compared with previous studies, why this method should be applied? And do these intermediates that the authors observed truly affect the riboswitch conformation and biological function in vivo?

Thank you very much for this important point. We realized that we did not clearly summarize our findings regarding the mechanical insights we gathered in a manner uniquely enabled by our approach. We have revised the manuscript accordingly.

There are several studies which identified pause sites from transcription termination experiments for riboswitches^{13,17,18,29,48} (and data not shown). Some pauses may have important regulatory effects, as demonstrated for other riboswitches. However, it is difficult to dissect the cause-and-effect relationship because changing pause sites requires either mutation of DNA templates, which can have secondary effects on folding (*i.e.*, mutating the terminator hairpin), or addition of proteins that modulate transcription rates (*i.e.*, Nus proteins), which also affect other transcription dynamics. We introduce an alternative approach to separately address the effect of pausing without changing RNA sequences or introducing transcription elongation factors and showed that pausing at a select position strongly influence riboswitching outcomes by allowing enough time for a key structural element to fold. To our

knowledge, this is the first time such mechanistic dissection has been performed in literature. To address this point, the relevant section now reads:

“VF assays also provided unique insights into the effect of various pause sites. In comparison, conventional RNAP-based methods have difficulties in assessing such effects because perturbing pause sites requires either mutation of DNA templates, which can have secondary effects on folding (*i.e.*, mutating the terminator hairpin), or addition of proteins that modulate transcription rates (*i.e.*, Nus proteins), which also affect other transcription dynamics. Therefore, it is challenging to probe the effect of specific pause sites in RNAP-based cotranscriptional folding experiments. Using VF assays, we found that using the pause mimics, *i.e.*, initiating unwinding after a slow folding element, promotes riboswitch activation at limiting ZMP concentrations, similar to the effect of slower unwinding speeds.”

Regarding the intermediates we observed, only one other study¹³ has investigated ZTP riboswitch folding using high-throughput sequencing methods and supports the intermediates described herein. However, using our method, we observed in real time that a subpopulation of folding RNAs occupying structure intermediates, which were previously not known to be functionally relevant for this RNA. Our approach circumvented the challenge of attaching fluorophore on “nascent RNA chains” encountered in RNAP-based smFRET cotranscriptional riboswitch folding studies. This advance made smFRET tools more applicable in riboswitch studies, offering the inherent ability to discern folding heterogeneity and subpopulations. In this sense, our approach has unique advantages in identifying these intermediates over other existing methods. Having seen such an effect, a future step would be to probe their function *in vivo*. However, we believe those experiments are beyond the scope of this manuscript. To address this point, the relevant section now reads:

“The findings here highlight the advantages of single-molecule approaches, in which folding heterogeneity can be directly observed in real time. VF assays circumvented the challenge of attaching fluorophore on “nascent RNA chains” encountered in RNAP-based smFRET cotranscriptional riboswitch folding studies, and thus making it possible to observe a much wider range of folding dynamics in real time. Future *in vivo* structural mapping studies can further validate the existence and roles of these intermediates.”

2) Related with the comment 1), do the folding dynamics revealed in this study reflect *in vivo* situation? If so, similar intermediates may also be found by *in vivo* SHAPE-map or related approaches that probe RNA structures. In other words, RepX-based RNA unwinding approach is very interesting, but it is unclear whether this system really reflects the situation of naturally occurring riboswitches or not.

To our knowledge, an *in vivo* structural map of this riboswitch is currently lacking. In general, structural mappings *in vivo* only look at steady-state RNA structures, most likely after RNAP has already gone through^{49,50}. Therefore, transient intermediates would be missed. Unlike such *in vivo* experiments, our studies monitored the folding heterogeneity in real time. The intermediates we observed with our RepX-based assays agree with the recent results from a RNAP-based *in vitro* SHAPE map¹³. These results, as the reviewer suggests, motivate attempting the *in vivo* experiments in the future. To address this point, the relevant section now reads:

“Future *in vivo* structural mapping studies can further validate the existence and roles of these intermediates.”

3) The generality of this approach is unclear. The authors verified the fluorophore labeling did not affect ligand-RNA binding so much. But it is unclear whether each RNA sequences used in this study show similar tendencies. Also, the difference between the helicase and RNAP may affect the parameters of RNA kinetics, transcription speed, and RNA folding and ligand binding. Can this unwinding method be applied to investigate other types of riboswitches? To show the generality of the method, the authors should test their approaches to other riboswitches (or RNA structures).

We believe this method is broadly applicable to many types of RNA and even DNA structures. In ref. 23, we applied this method to study the folding behavior of a Twister ribozyme. The same method has also been used to study RNA aptamers (e.g., Spinach and Mango) and telomeric DNA sequences to mimic co-reverse transcriptional telomeric DNA folding^{24,35}. The challenge of fluorophore labeling is faced in any type of single-molecule fluorescence experiment and is not particularly trickier in this method. To address this point, the relevant section now reads:

“As many nucleic acid folding processes are kinetically controlled including those involving ribonucleic protein complexes^{39,40}, we believe this method is broadly applicable to many types of RNA and even DNA structures, as it has been demonstrated in ribozymes (e.g., Twister²³), RNA aptamers (e.g., Spinach and Mango²⁴) and even telomeric DNA sequences in an experimental format that mimics co-reverse transcriptional telomeric DNA folding³⁵, and to the studies of nucleic acids-protein complex assemblies.”

Minor points

1) In Figure1b legend, the abbreviation VF is not explained.

We have fixed this. The relevant part of the legend now reads: “(b) Cartoon depicting the vectorial folding (VF) assay, in which an engineered helicase Rep-X (orange and green) releases the riboswitch RNA (black) in a 5'-to-3' direction by translocating on the cDNA strand (grey) and unwinding in a 3'-to-5' direction.”

2) In Figure2a, the blue line should be explained.

We have fixed this. The legend now reads: “(a) The 75-nt ZTP riboswitch aptamer domain (Δ term) was thermally refolded at 0 mM ZMP and then incubated with different concentrations of ZMP. Gaussian fitting with global constraints was used to determine the relative population of the ZMP-bound (magenta) vs. ZMP-unbound (green) aptamer. The zero- E_{FRET} state (blue) was also fit to account for a small fraction (~5 %) of annealed RNA due to remaining splint ssDNA from sample preparation.”

3) In Figure3b, the number above the gel image is not described in legend. The number of plots in quantification is 9, but the number of gel image is 10.

We have fixed this by adding a black bar to the figure and revising the legend. The relevant part of legend now reads: "Bulk single-round transcription termination experiments of ZTP riboswitch are shown in the presence of 0.2 mM NTPs and varying amounts of ZMP (0-4 mM)."

4) In figure6, the attachment of RNA model (structure) images is helpful for understanding the kinetic changes of RNA folding.

The secondary structure is present in Figure 1a. To avoid reproducing this structure and for clarification, we have added the following sentence to the relevant results section: "Using the measured average Rep-X unwinding speed, the progress of Rep-X unwinding through the ZTP riboswitch structural elements (Fig. 1a) vs. time can be estimated (Fig. 6b)."

48. Steinert, H., Sochor, F., Wacker, A., Buck, J., Helmling, C., Hiller, F., Keyhani, S., Noeske, J., Grimm, S., Rudolph, M.M., *et al.* (2017). Pausing guides RNA folding to populate transiently stable RNA structures for riboswitch-based transcription regulation. *eLife* 6.

49. Ding, Y., Tang, Y., Kwok, C.K., Zhang, Y., Bevilacqua, P.C., and Assmann, S.M. (2014). *In vivo* genome-wide profiling of RNA secondary structure reveals novel regulatory features. *Nature* 505, 696-700.

50. Dar, D., Shamir, M., Mellin, J.R., Koutero, M., Stern-Ginossar, N., Cossart, P., and Sorek, R. (2016). Term-seq reveals abundant ribo-regulation of antibiotics resistance in bacteria. *Science* 352, aad9822.

Reviewer #2 (Remarks to the Author):

In this manuscript, Hua and Jones, et al. investigate the co-transcriptional RNA folding mechanism of the *F. ulcerans* ZTP riboswitch at single-molecule resolution using a vectorial RNA folding assay that was previously developed by the Ha lab. The vectorial folding assay uses a processive helicase (Rep-X) to sequentially release a fluorophore-labeled RNA from an RNA/DNA heteroduplex thereby mimicking co-transcriptional RNA folding. This approach enables vectorial RNA folding kinetics to be quantified by single-molecule FRET, which is not easily achieved using RNA polymerase (RNAP)-based assays. This work presents the first application of vectorial RNA folding to study riboswitch RNA folding kinetics.

Under equilibrium conditions, the authors observe that the ZTP aptamer domain folds into distinct conformations in the absence and presence of ZMP. These conformations correspond to ZMP-unbound and ZMP-bound states. If the transcription terminator domain is present, the ZMP-bound aptamer-folded state can only be accessed by vectorial folding. This suggests that *F. ulcerans* ZTP riboswitch folding is kinetically controlled and is consistent with the authors' measurements from corresponding bulk in vitro transcription assays. The authors then perform a series of experiments showing that unwinding velocity (which simulates transcription speed), pre-equilibration of defined RNA segments (which simulates transcription pausing), and terminator hairpin stability all contribute to RNA folding outcomes. Together these data strongly support a kinetic folding mechanism. Lastly, the authors measure ZTP riboswitch folding in real-time and are able to quantify differences in terminator folding rates between the WT riboswitch and a mutant with a disrupted terminator stem. Notably, the 81-83A mutant only destabilizes terminator folding under vectorial conditions. This suggests that the deficiency observed for the 81-83A terminator in bulk transcription assays is the result of a kinetic RNA folding defect, rather than terminator hairpin destabilization.

The authors' experiments are convincing, their conclusions are well-supported, and the findings are presented clearly. Both the application of the vectorial RNA folding assay to study a riboswitch RNA and the authors mechanistic findings are significant advances and will be of broad interest. Experimentally, this work is suitable for publication in *Nature Communications* as-is. However, the authors should consider the points below, which may improve the manuscript.

Major comments:

My only significant suggestion is that the authors present a fairer comparison of their vectorial folding assay to alternative measurement approaches in the introduction. The authors accurately identify the limitations of bulk approaches (structure probing, NMR, and fluorescently labeled ligands) and alternative single-molecule approaches (optical tweezers) for measuring riboswitch RNA folding; the manuscript would be improved if in addition to describing the advantages of their own methodology they also acknowledge its limitations (which, to be clear, do not impact the validity of the presented work).

We have revised the discussion in the manuscript to reflect this. The relevant section now reads:

“A potential drawback of VF assays is that replacing RNAP with a helicase cannot recapitulate some RNAP-specific interactions with nascent chains, though it helps to more clearly dissect the contributions of other key kinetic parameters. In addition, we cannot rule out the effect of potential contacts between the unwound ssDNA and RNA chains, though comparisons made with bulk kinetics and steady-state single-molecule kinetics here and elsewhere²³ suggest that such an effect is not significant at least in the systems we have examined so far. These interactions between ssDNA and RNA chains can also occur with RNAP, such as those in R-loop structures³⁶. Thirdly, the current unwinding yield of VF assays is ~30%, which is primarily due to the fact that not every construct has a helicase bound at the helicase concentrations we used,

and some helicase dissociates before it begins unwinding²³. Although 30% yield should be sufficient for many studies, we anticipate that future developments will improve the yield.”

While I agree that simulating co-transcriptional RNA folding by helicase unwinding has some distinct advantages relative to RNAP-based assays, the absence of polymerase characteristics can also be viewed as a limitation because it is well-established that interactions between nascent RNA and a transcribing RNAP have quantifiable effects on RNA folding. Additionally, the suggestion that the vectorial folding assay is less susceptible to RNA mutations than RNAP-based assays (lines 83-87) does not seem correct because the labelled RNA is assembled from a transcript synthesized by an RNAP (which should have the same error rate as RNAP-based assays) and a chemically synthesized RNA oligonucleotide (which likely has a worse error rate than typically used RNAPs); unless the authors have some other justification for this statement it should probably be removed.

The first part of the question is addressed as above. When referring to mutations, we were describing designed mutations for the purpose of probing function (rather than random mutations due to the intrinsic polymerase error rate). As we see that our point was unclear, we have explicitly stated this point:

“Another issue that arises when using RNAP to examine cotranscriptional RNA folding is that experimental mutations to the DNA template, hence changing the RNA sequence, can affect both RNA folding and interactions between RNAP and nucleic acid substrates. For example, mutations to the terminator hairpin can result in changes in its folding and its interactions with RNAP that a single readout of transcription termination efficiency may be compounded by both effects.”

Lastly, how does the emergence of ssDNA from Rep-X as the RNA/DNA heteroduplex is unwound impact RNA folding? Presumably the ssDNA can interact with the ssRNA at some frequency given that these nucleic acids are complementary and in close proximity. If this is a relevant consideration, the authors should state how it is addressed.

We have considered this point in our previous study²³. In that study we designed a “trap” oligo to bind and sequester the ssDNA, and we found no difference in unwinding yields or folding behaviors whether the “trap” oligo was included or not during unwinding. A possible explanation is that the ssDNA can also “fold” and therefore sequester itself from interfering with RNA folding, although we cannot conclude that it makes no contact at all with the released RNA. Similar interactions can also occur with RNAP, such as those in R-loop structures (although a complementary DNA strand lowers the overall likelihood of this event). Overall, our assay faithfully recapitulated the behaviors and trends of RNA folding when benchmarked against RNAP-based approaches, suggesting it is unlikely to cause an artifact in the study.

To clarify this point, we included a statement in the manuscript that reads:

“In addition, we cannot rule out the effect of potential contacts between the unwound ssDNA and RNA chains, though comparisons made with bulk kinetics and steady-state single-molecule kinetics here and elsewhere²³ suggest that such an effect is not significant at least in the systems

we have examined so far. These interactions between ssDNA and RNA chains can also occur with RNAP, such as those in R-loop structures³⁶.”

Minor Comments:

1. In Figure 1B it would be helpful to have labels indicating the ZMP condition on the plots.

We are not sure what this is referring to. Figure 1B is a cartoon overview in which ZMP concentrations should not be shown.

2. In Figure 3B, what do the numbers above the gel lanes indicate? Unless these numbers have a purpose, it might be better if the authors use a black triangle and indicate ZMP concentration is increasing from left to right.

We have fixed this as suggested by the reviewer. The figure now includes a black triangle, and the relevant section of the legend now reads: “Bulk single-round transcription termination experiments of ZTP riboswitch are shown in the presence of 0.2 mM NTPs and varying amounts of ZMP (0-4 mM).”

3. On line 296, should ‘absence of trimethoprim’ be ‘presence of trimethoprim’?

We have fixed this. The manuscript now reads: “The total pool of ZMP and ZTP ranges from ~0.2 mM to ~1.3 mM in *E. coli* depending on the presence of trimethoprim⁴, so even basal levels of Z nucleotides should be sufficient to activate the ZTP riboswitch if transcription is hindered at the same time.”

4. In Supplementary Figure 2E the authors should consider showing the full y-axis range on both plots so that readers can directly compare the 10 and 200 micromolar NTP conditions. This is important since comparison of these results supports the authors finding of a kinetic folding mechanism.

We have fixed this and placed them on the same scale such that y-axis \in [0-100].

5. In Supplementary Figure 3, it would be helpful if panels E, G, and H had better annotations.

We have fixed this, including annotations to the figure as well revisions to the legend. The legend now reads: “(e) The percentage of aptamer fold obtained by VF by Rep-X for WT (black) and terminator mutant 81-83A (blue) in the presence of ZMP (*mean \pm s.e.m.*, *n* = 3-4). (f) The E_{FRET} histograms of 81-83A thermally refolded at 0 mM (left) and 1 mM ZMP (right). To distinguish the terminator from the ZMP-unbound aptamer, 81-83A was imaged at 1 mM ZMP after being refolded at different ZMP concentrations. Gaussian fitting with global constraints was used to determine the relative population of ssDNA-bound RNA (blue, see also Fig. 2a) and refolded (green). (g) E_{FRET} histograms of terminator mutant 92-94A, thermally refolded in the absence of ZMP and imaged at 0 mM (top) and 1 mM ZMP (bottom).

Gaussian fitting with global constraints was used to determine the relative population of free (green) and ZMP-bound (magenta) RNA. (h) E_{FRET} histograms of terminator mutant 81-83,92-94A, thermally refolded in the absence of ZMP and imaged at 0 mM (top) and 1 mM ZMP (bottom). Gaussian fitting with global constraints was used to determine the relative population of free (green) and ZMP-bound (magenta) RNA.”

Reviewer #3 (Remarks to the Author):

In this paper, Hua et al. used a single-molecule vectorial folding assay to mimic the cotranscriptional folding of a riboswitch, taking advantage of an engineered superhelicase developed by the same group that sequentially releases single-stranded RNA from a heteroduplex at a speed comparable to in vivo transcription. This assay allowed them to monitor the riboswitch folding behavior under different kinetically controlled conditions, including unwinding speed, position-specific pausing, and terminator strength. The results suggest that a longer time window before terminator formation promotes ligand binding to the riboswitch, leading to its activation.

This elegant work overcomes several limitations of previous studies of cotranscriptional folding, contributes important insights into the kinetic control of riboswitch folding, and can be applied to many other folding systems. The work involves a variety of cleverly designed assays and constructs, showcasing the strength of single-molecule fluorescence spectroscopy in dissecting dynamic biological processes. This paper will be of general interest to researchers in the fields of single-molecule biophysics and gene regulation.

The main concern on the paper is regarding its data presentation. Although the introduction and discussion are nicely written, the results and figures are currently presented in a way that is difficult to follow. Some interpretations also need to be clarified. These issues should be addressed before publication.

Specific comments:

(1) The study uses many different constructs. They should be clearly depicted in the figure panels (e.g., Fig 2A, 3A, 4A, 4D, 4E, S2, S3D-S3H).

Most samples we used are closely related to the WT, with mutations at specific sites, or shorter cDNAs to mimic pauses. To address this point, we included descriptions at the top of the histograms in panels for Figs. 2, 3 and 4 and Figs. S2 and S3.

(2) The zero FRET state in Fig 2D and 3A was interpreted as the starting substrate conformation for the vectorial folding assay. Why is its occupancy much higher for the WT substrate than the Δ_{term} substrate? What is the zero FRET state in the thermal folding assay in Fig 2A? Are those donor-only molecules? If so, how were they distinguished from the conformationally extended molecules?

For the WT construct (Fig. 3A), we used the typical “single-turnover” unwinding assay, which has the unwinding yield of about 30%. The advantage of single-turnover assays is that all helicases can be synchronized to the addition of ATP. For the Δ_{term} construct (Fig. 2D), we employed a “multiple-turnover” assay, which allows helicase rebinding and multiple rounds of unwinding, to maximize the unwinding yield.

As for Fig. 2A, we have included an explanation in the legend.

(3) Judging from Fig 3A, a large fraction of molecules did not respond to helicase unwinding. What is the explanation for such a low efficiency?

We discussed this in our previous paper²³. Briefly, not every construct has a helicase bound at the helicase concentrations we used, and some helicase dissociates before it begins unwinding.

(4) Are the green (0.2 FRET) and magenta (0.5 FRET) peaks in Fig 3 considered to be in the same structural configurations as the green (0.25 FRET) and magenta (0.6 FRET) peaks in Fig 2? If not, they should be shown in different colors.

They represent the same structures.

(5) Sample FRET trajectories should be added to Fig 2D, 3 and 4, as they are important for differentiating static heterogeneity from dynamic heterogeneity among FRET states. For example, in Fig 3A, are there interconversions between the 0.2 and 0.5 FRET states for the same molecule? This would provide support to the statement that “this ligand-bound conformation is stable during our measurements” (Line 150).

We now included example trajectories for Fig. 3A (WT 0.01 mM ZMP in Supplementary Fig. 6F). Example trajectories for Fig. 3A (at 0 and 1 mM ZMP) are already presented in Fig. 6A and 6B and Supplementary Fig. 6B and 6D. We also presented evidence for the quoted statement in Supplementary Fig. 2B and 2C.

(6) I appreciate the multiple ways to calculate the unwinding rate in Fig 5. However, they largely arrived at the same results. I am not sure if they all need to be in the main figure. Furthermore, the distributions

for the unwinding rate in Fig 5 are all quite broad, suggesting substantial molecule-to-molecule heterogeneity. This should be taken into consideration when calculating the unwinding time later in the manuscript. For example, the fast and slow molecules may take different folding paths in the real-time experiments in Fig 6.

We prefer to keep the data in the main figures, because they present novel method developments that are of interest to a broader audience who may not be interested in this particular riboswitch *per se*.

As for the unwinding speed heterogeneity, this is an excellent point. We observed this unwinding speed heterogeneity in our previous studies²². We believe the broad rate distribution here is indeed fascinating, and the method was designed to measure molecule-to-molecule rate heterogeneity to allow for the observation of different folding paths in real time. In pilot studies, we observed a correlation between the PIFE width and outcome such that wider PIFE signals (*i.e.*, slower unwinding) was correlated with higher FRET states (*i.e.*, more riboswitch activation). However, after further experimentation, no such correlation was observed. It is possible that a much larger sample size (real-time folding experiments at several ZMP concentrations) would be necessary to confirm such a correlation.

(7) The real-time unwinding experiment in Fig 6 is the most exciting part of this study. However, the data are not sufficiently analyzed. How many FRET states were observed in the trajectories? Judging from the overlaid traces, there seem to be at least three states (~ 0.2 , ~ 0.5 , and ~ 0.8). What do they each correspond to? How do their lifetimes differ between the WT and 81-83A mutant? In the abstract the authors mentioned “an intermediate responsible for delayed terminator formation.” Which state does it refer to? Ideally, the results should be presented in alternative folding pathways, each with a representative trajectory and illustration. While I understand that the statistics are low, which may prevent the authors from making more quantitative conclusions, a more in-depth analysis would still help to better understand the complexity of the system. At a minimum, the “delayed formation of the mutant terminator” should be supported by a more rigorous quantitation.

The ~ 0.2 E_{FRET} state is the terminated state, the ~ 0.5 E_{FRET} state is the aptamer or aptamer-like state, and the ~ 0.8 E_{FRET} state is a novel state that could be either on-pathway (*e.g.*, intermediates of strand displacement) or off-pathway. The percentage of trajectories that showed prolonged stay (≥ 3 s) in the high E_{FRET} states (≥ 0.7) was used as a measure for the delayed terminator formation. We also included E_{FRET} histograms of WT and 81-83A at different time points after PIFE (*i.e.*, 0 s, 2 s and 8 s).

(8) A model figure summarizing the folding behaviors under different kinetic conditions would be very helpful. Can the different FRET states be related to specific predicted structures of the ZTP riboswitch?

We have included a model as Figure S6. The model includes the RNA folding intermediate and terminator hairpin, noting the effects of unwinding speed, pauses, and terminator stability.

(9) There are a number of mislabels and omissions, especially in the supplementary figures. To name a few:

Fig S2C: circles were not labeled.

The legend now reads: “(c) The remaining percentage of aptamer vs. the post-ATP addition time in VF assays. Each circle represents an individual experiment at the specified time point. The percentage of aptamer is calculated as the ratio of the aptamer population (magenta) over the sum of the terminator (green) and aptamer populations.”

Fig S3D: should be 0 mM and 0.1 mM ZMP rather than ATP.

The figure was updated.

Fig S3E: what do the blue and black circles stand for?

The figure was updated, and the legend now reads: “(e) The percentage of aptamer fold obtained by VF by Rep-X for WT (black) and terminator mutant 81-83A (blue) in the presence of ZMP (*mean ± s.e.m.*, $n = 3-4$).”

Fig S3 legend: the 87C variant was not described anywhere in the text or figures.

This data was removed, and the figure was updated. The legend now reads: “(c) T50 values are shown for the terminator variants in **b**. Values are *mean ± s.d.* with $n \geq 3$ independent experiments.”

Fig S4A: it was not explained how the traces were segmented to calculate the unwinding times.

We described the analysis in the method section. The legend now reads: “(a) Individual single-molecule trajectory (top) and histogram of all dropoff unwinding events (bottom) for Rep-X unwinding of indicated segments of the ZTP riboswitch (see Online Methods for Δt calculation).”

REVIEWERS' COMMENTS:

Reviewer #1 (Remarks to the Author):

The revised manuscript satisfactory responded to my previous concerns and I believe it is now ready for publication.

Reviewer #2 (Remarks to the Author):

The authors have fully addressed all of my comments. The manuscript is an important contribution to the field and I recommend that it be accepted for publication.

Reviewer #3 (Remarks to the Author):

The authors have adequately addressed my concerns in the revised manuscript. I now recommend its publication at Nature Communications.

No additional comments were raised by reviewers in this round of revision

REVIEWERS' COMMENTS:

Reviewer #1 (Remarks to the Author):

The revised manuscript satisfactory responded to my previous concerns and I believe it is now ready for publication.

Reviewer #2 (Remarks to the Author):

The authors have fully addressed all of my comments. The manuscript is an important contribution to the field and I recommend that it be accepted for publication.

Reviewer #3 (Remarks to the Author):

The authors have adequately addressed my concerns in the revised manuscript. I now recommend its publication at Nature Communications.